# Geometrical Optimal Navigation and Path Planning—Bridging Theory, Algorithms, and Applications

**DOI:** 10.3390/s25226874

**Published:** 2025-11-11

**Authors:** Hedieh Jafarpourdavatgar, Samaneh Alsadat Saeedinia, Mahsa Mohaghegh

**Affiliations:** 1Control Engineering Department, Amirkabir University, Tehran 1591634311, Iran; hedieh.jafarpourdavatgar@gmail.com; 2Control Electrical Engineering Department, Iran University of Science of Technology, Tehran 1684613114, Iran; s_saeedinia@alumni.iust.ac.ir; 3Faculty of Design and Creative Technologies, Auckland University of Technology, Auckland 1142, New Zealand

**Keywords:** geometric navigation, path planning, optimization algorithms, autonomous systems, collision-free navigation, dynamic environments

## Abstract

Autonomous systems, such as self-driving cars, surgical robots, and space rovers, require efficient and collision-free navigation in dynamic environments. Geometric optimal navigation and path planning have become critical research areas, combining geometry, optimization, and machine learning to address these challenges. This paper systematically reviews state-of-the-art methodologies in geometric navigation and path planning, with a focus on integrating advanced geometric principles, optimization techniques, and machine learning algorithms. It examines recent advancements in continuous optimization, real-time adaptability, and learning-based strategies, which enable robots to navigate dynamic environments, avoid moving obstacles, and optimize trajectories under complex constraints. The study identifies several unresolved challenges in the field, including scalability in high-dimensional spaces, real-time computation for dynamic environments, and the integration of perception systems for accurate environment modeling. Additionally, ethical and safety concerns in human–robot interactions are highlighted as critical issues for real-world deployment. The paper provides a comprehensive framework for addressing these challenges, bridging the gap between classical algorithms and modern techniques. By emphasizing recent advancements and unresolved challenges, this work contributes to the broader understanding of geometric optimal navigation and path planning. The insights presented here aim to inspire future research and foster the development of more robust, efficient, and intelligent navigation systems. This survey not only highlights the novelty of integrating geometry, optimization, and machine learning but also provides a roadmap for addressing critical issues in the field, paving the way for the next generation of autonomous systems.

## 1. Introduction

The automated industry is rapidly evolving, with robots being increasingly deployed in diverse applications such as manufacturing, healthcare, transportation, and smart homes [1,2]. To operate safely and efficiently in these environments, robots must navigate autonomously while avoiding obstacles, making path planning and motion planning critical components of robotics. Path planning involves generating a collision-free trajectory that guides a robot from an initial state to a desired goal. This requires accurate environment modeling to understand the spatial structure and relationships between locations. Three primary approaches are used to link the environment and navigation strategy: *Geometric*, *Topological*, and *Semantic*.

*Geometric methods* focus on guiding the robot from a start point to a goal based on map information, often leveraging mathematical representations of the environment [3]. These methods are widely used due to their simplicity and efficiency in static environments. *Topological methods* represent the environment as a graph, enabling decision-making that mimics human-like navigation [4]. *Semantic approaches* use logical representations of the environment, incorporating human cognitive processes to infer navigation strategies [5]. While topological and semantic methods are gaining attention for their adaptability, geometric approaches remain highly relevant, especially when combined with optimization techniques [6].

This paper provides a systematic overview of geometric optimal navigation and path planning, focusing on the integration of classical geometric methods with modern optimization and machine learning techniques. We explore recent advancements in continuous optimization, real-time adaptability, and learning-based strategies, which enable robots to navigate complex environments, avoid moving obstacles, and optimize trajectories under constraints. Additionally, we highlight unresolved challenges, including scalability in high-dimensional spaces, real-time computation for dynamic environments, and the integration of perception systems for accurate environment modeling. Safety concerns in human–robot interactions are also discussed as critical issues for real-world deployment [7].

## 2. Navigation Methods

Path planning algorithms are essential for enabling autonomous robots to navigate complex environments efficiently. These algorithms can be broadly categorized into three primary approaches: **classical**, **reactive**, and **hybrid**. Each category addresses distinct challenges in path planning, offering unique advantages and limitations depending on the application and environment.

### 2.1. Classical Path Planning Methods

Classical path planning methods are rooted in geometric and mathematical approaches, relying on map reconstruction and environmental geometry to construct a navigable graph. These methods typically focus on finding the shortest or safest path between a start point and a target destination. Key techniques within this category include the following:**Roadmap Methods**: These methods create a graph representation of the environment, connecting obstacles and the target point. Examples include the following:–**Visibility Graph**: Generates shorter paths by connecting visible vertices of obstacles, optimizing for path length [8].–**Voronoi Diagram**: Prioritizes safety by creating paths that maximize the distance from obstacles, suitable for environments where collision avoidance is critical [9].**Cell Decomposition**: This approach divides the workspace into smaller cells, enabling the robot to navigate through collision-free regions. Notable examples include the following:–**Lozano-Perez’s C-Space Decomposition (1983)**: Treats the robot as a C-shaped object and subdivides the environment into cells to identify feasible paths [10].–**Grid-Based Methods**: Popularized by Hachour (2008), this technique discretizes the environment into a grid, simplifying path planning through cell-by-cell exploration [11].**Potential Field Method**: Models the robot as a particle influenced by artificial potential fields. Attractive potentials guide the robot toward the goal, while repulsive potentials push it away from obstacles. This method ensures smooth navigation but can suffer from local minima issues [12].**Mathematical Programming**: Formulates path planning as an optimization problem, treating obstacle avoidance as a set of inequalities. The goal is to minimize a scalar quantity (e.g., path length or energy consumption) while finding a feasible curve between the start and target points [8].

Classical methods excel in static environments with well-defined maps but often struggle in dynamic settings due to their reliance on precomputed maps and lack of real-time adaptability.

### 2.2. Reactive Path Planning Methods

Reactive methods prioritize real-time decision-making by relying on sensory data rather than preconstructed maps. These approaches are well-suited for dynamic environments where obstacles and conditions change unpredictably. Key reactive techniques include the following:**Subsumption Architecture**: Organizes robot behaviors into hierarchical layers, with higher-level actions overriding lower-level ones. This structure enables quick responses to environmental changes but may lack global optimization [13].**Motor Schemas**: Generates output vectors for distinct behaviors (e.g., obstacle avoidance, goal seeking), which are combined through vector summation to determine the robot’s overall response. This approach allows for flexible and adaptive navigation [14].

While reactive methods excel in dynamic environments and ensure collision avoidance, they do not guarantee optimal paths and may result in inefficient or oscillatory motion.

### 2.3. Hybrid Path Planning Methods

Traditional (classical) methods, often categorized as global navigation techniques, rely heavily on map reconstruction. While they ensure optimal or near-optimal paths, their computational burden increases significantly with larger or more complex environments [15]. Additionally, they may fail to handle dynamic obstacles effectively. Reactive methods, on the other hand, excel in dynamic settings but lack global optimization, potentially leading to sub-optimal paths [16]. Hybrid methods combine the strengths of classical and reactive approaches, addressing the limitations of each. By integrating global planning with real-time adaptability, hybrid methods offer robust solutions for both static and dynamic environments. These methods can be categorized as follows:**Managerial Approaches**: Use a high-level planner to generate global paths while employing reactive strategies for local obstacle avoidance [17].**State Hierarchies**: Organize navigation tasks into hierarchical states, enabling seamless transitions between global and local planning [18].**Model-Oriented Styles**: Incorporate environmental models to enhance decision-making, balancing long-term planning with real-time adjustments [19].

Hybrid methods are particularly effective in complex environments, as they leverage the precision of classical methods and the adaptability of reactive strategies. However, they require careful design to balance computational efficiency and real-time performance.

Beyond the classification of methods into classic, reactive, and hybrid, three critical dimensions deeply influence system effectiveness: scalability, real-time performance, and adaptability. Scalability addresses how well a navigation system maintains efficiency and responsiveness as the operational environment grows in size or complexity, or when multiple robots coordinate. Real-time performance ensures that all sensing, planning, and control computations are completed within strict timing constraints to enable prompt and reliable responses. Adaptability captures the capability of robotic systems to dynamically perceive and react to changing, unpredictable environments. Together, these factors critically shape the design and practical deployment of navigation strategies [20,21,22]. The choice and design of navigation algorithms play a pivotal role in addressing the intertwined challenges of scalability, adaptability, and real-time performance in robotic systems. Scalable algorithms efficiently manage increasing complexity by employing hierarchical structures, decentralized computations, or approximate methods that reduce computational overhead while preserving critical decision accuracy. Adaptability is achieved through algorithms capable of continuous environment sensing and dynamic re-planning, often integrating learning-based or predictive components that can respond flexibly to unforeseen changes. Ensuring real-time performance requires that these computations, including sensing, perception, planning, and control, are optimized for low latency and deterministic execution within strict timing constraints. By carefully balancing these factors in algorithm design—often through hybrid combinations of reactive and planning methods—navigation systems can robustly operate in complex, dynamic environments with guaranteed responsiveness and safety [21,23,24,25]. Figure 1 indicates the classification of path planning algorithms.

## 3. Optimization Criteria in Geometric Trajectory Planning

In geometric trajectory planning, the primary and most observable parameter is the determination of the shortest possible trajectory. However, while trajectory length is a critical factor, it should not be the sole criterion for optimization. A comprehensive approach must consider additional factors such as trajectory smoothness, time efficiency, energy consumption, and potential hazards, including collisions or environmental disturbances like wind resistance. Furthermore, the robot’s specific task and objectives often necessitate the inclusion of other pertinent models and metrics to ensure optimal performance.

It is important to note that multiple optimization criteria often exhibit inherent interdependencies. The way these interdependencies are addressed depends largely on the priorities assigned to the navigation problem and the sensitivity of the scenario. For instance, in safety-critical applications, collision avoidance may dominate, while in energy-constrained missions, energy efficiency might take precedence. Common approaches include weighted-sum formulations, hierarchical prioritization, or treating certain objectives as hard constraints while optimizing others. While a fully generalized treatment of these interdependencies is beyond the scope of this work, this overview illustrates how diverse objectives can be balanced to achieve feasible, efficient, and robust navigation outcomes [26,27].

### 3.1. Key Optimization Criteria

The following criteria are essential for evaluating and optimizing trajectories in geometric path planning:**Trajectory Length**: Minimizing the path length is often the primary objective, as it directly impacts the robot’s efficiency and resource utilization. Shorter trajectories reduce travel time and energy consumption, making them ideal for many applications [8].**Trajectory Smoothness**: Smooth trajectories are crucial for ensuring stable and efficient robot motion. Abrupt changes in direction or velocity can lead to mechanical stress, increased energy consumption, and reduced accuracy. Smoothness is often quantified using curvature and jerk metrics [28].**Time Efficiency**: Time-optimal trajectories are critical in applications where speed is a priority, such as in industrial automation or search-and-rescue operations. Time efficiency is closely tied to the robot’s velocity profile and acceleration limits [29].**Energy Consumption**: Energy-efficient trajectories are vital for battery-powered robots or systems operating in energy-constrained environments. Optimizing energy usage involves minimizing unnecessary acceleration, deceleration, and idling [30].**Collision Avoidance**: Ensuring collision-free trajectories is a fundamental requirement in any navigation task. This involves not only avoiding static obstacles but also dynamically adapting to moving obstacles in real-time [12].**Environmental Factors**: External conditions such as wind resistance, terrain roughness, or fluid dynamics (in underwater or aerial robots) can significantly impact trajectory planning. These factors must be modeled and accounted for to ensure robust performance [31].

### 3.2. Path Length as an Optimization Criterion

Path length is one of the most critical parameters in the optimization of mobile robot path planning. It is frequently employed as a primary criterion for evaluating and optimizing trajectories. The length of a path is typically calculated by summing the distances traveled by the robot at each time step as it moves from its initial position to the target destination. Mathematically, path length can be determined using algebraic norms, with the Euclidean norm (or $L_{2}$ norm) being one of the most widely used distance metrics. The general definition of the $L_{p}$ norm, denoted as ${∥\cdot ∥}_{p}$, between two points in an *n*-dimensional space is given by(1)$${∥x∥}_{p}={\left(\sum_{i=1}^{n}{\left|x_{i}\right|}^{p}\right)}^{1/p},$$
where $x=(x_{1},x_{2},\ldots ,x_{n})$ is a vector representing the coordinates of a point, and *p* is the order of the norm. For path planning in three-dimensional (3D) space, n=3, and the coordinates *x*, *y*, and *z* correspond to the Cartesian axes.

The Euclidean norm ($L_{2}$ norm) is a special case of the $L_{p}$ norm where p=2. It is defined as follows:(2)$${∥x∥}_{2}=\sqrt{x^{2}+y^{2}+z^{2}}.$$


**Path Length Calculation**


To calculate the total length of a path, the Euclidean norm is applied to each segment of the path. For a path consisting of *m* points, the total path length $L_{k}$ for the *k*-th path is given by:(3)$$L_{k}=\sum_{l=1}^{m-1}{∥p_{l+1}-p_{l}∥}_{2},$$
where $p_{l}=\left(x_{l},y_{l},z_{l}\right)$ represents the coordinates of the *l*-th point on the path, and *m* is the total number of points, including the starting and target points. To illustrate this concept, consider Figure 2, which depicts two possible paths from an initial state to a target destination. Path 1 consists of 3 intermediate points (m=4), while Path 2 consists of 2 intermediate points (m=3). Using Equation (Equation 3), the total path lengths $L_{1}$ and $L_{2}$ for Path 1 and Path 2, respectively, can be calculated. Typically, the path with the shorter length is preferred, as it minimizes travel distance and, consequently, energy consumption and time expenditure [8].

While minimizing path length is a common objective in path planning, it is essential to consider other factors such as trajectory smoothness, collision avoidance, and energy efficiency. For instance, a shorter path may require sharper turns or higher acceleration, which can increase energy consumption and mechanical stress on the robot [29]. Therefore, path length should be optimized in conjunction with other criteria to achieve a balanced and efficient trajectory.

### 3.3. Path Smoothness in Robotic Navigation

The notion of path smoothness in navigation arises from the limitations imposed by the robot’s constrained angular velocity or the disparity in movement angles between its current intended state and the subsequent state (Figure 3) [32]. However, it is essential to acknowledge that the smoothness of a planned path is a complex outcome influenced by various factors, encompassing the robot’s dynamics and kinematics, as well as the dynamics associated with surrounding obstacles. Parameters such as angle, velocity, acceleration, and environmental dynamics collectively contribute to determining the displacement and regularity of a path. Neglecting elements such as obstacle avoidance, non-holonomic features, and speed limitations can result in a significant disparity between the intended course and the actual path followed by the robot [32].

The pursuit of a smooth and uninterrupted path is motivated by its capability to enable the robot to navigate without sudden and acute turns. To accomplish path smoothness, three primary approaches are commonly employed: interpolation [32,33], special curves [33,34], and optimization [35,36]. Interpolation algorithms, while aiming to generate regular paths, can encounter challenges such as high computational costs and the non-convergence issue known as Runge’s phenomenon [37]. Transition curves, connecting straight segments of a path with a curve, offer advantages like zero start curvature, tangential joining of circular curves, and a uniform rate of curvature change. However, they necessitate the tuning of control points and parameters and may incur high computational costs, proving less effective at higher speeds [38].


**Optimization-Based Path Smoothing**


One notably effective technique for achieving path smoothness is optimization-based path smoothing. This approach involves identifying the best path that satisfies various criteria, encompassing path length, safety, and energy consumption. Formulating the path design problem as an optimization issue allows for the development of algorithms that seek paths meeting these criteria. Recent research has extensively explored diverse optimization techniques for path design and trajectory optimization in both ground and aerial vehicles [39,40,41].

Figure 3 provides a visual comparison between non-smooth and smooth paths, clearly illustrating that the smooth path not only covers a shorter distance but also follows a feasible route in terms of the robot’s dynamics. Accordingly, Figure 3 illustrates that the angular difference is reduced in the smooth path when compared to the non-smooth path.


**Three-Dimensional Navigation**


Optimal strategies for navigating in three-dimensional environments have witnessed notable advancements, as a variety of approaches have been developed that incorporate transition curves to abide by kinematic and dynamic motion constraints. Transition curves play a crucial role in achieving the seamless integration of smooth paths within these navigation frameworks. The key lies in ensuring that these curves meet the intricate requirements imposed by the motion dynamics of the robot, thus guaranteeing a harmonious trajectory. The notion of path smoothing, which is essential for enhancing the efficiency of robotic systems in terms of navigation, is intricately intertwined with a set of motion constraints. This becomes particularly vital in the context of three-dimensional environments, where the robot must traverse complex spatial configurations. The utilization of transition curves serves as an advanced solution, contributing to the overall optimization of the robot’s trajectory. Recent studies have introduced advanced smoothness criteria to address the limitations of traditional methods. These include the following:**Curvature Continuity**: Ensuring that the curvature of the path is continuous, which is critical for high-speed navigation and dynamic environments [42].**Jerk Minimization**: Minimizing the rate of change of acceleration (jerk) to ensure smoother motion and reduce wear on the robot’s actuators [43].**Energy-Efficient Smoothing**: Optimizing paths to minimize energy consumption, which is particularly important for battery-operated robots [44].**Adaptive Smoothing**: Dynamically adjusting the smoothness of the path based on environmental changes and obstacle movements [45].


**Path Smoothing Methods**


Table 1 provides a summary of some of the well-established path smoothing methods, including recent advancements. These methods leverage transition curves and encompass a diverse array of techniques aimed at addressing motion constraints. The table also includes detailed descriptions of the parameters and variables used in the mathematical formulations.

This section has presented a holistic perspective on the integration of transition curves in optimal navigation methods, emphasizing the crucial consideration of both kinematic and dynamic constraints. By doing so, the goal is to contribute to the development of a comprehensive and efficient robotic navigation experience. In this context, optimization-based methods emerge as a key player, adept at striking a delicate balance between competing criteria.

Optimization-based methods, showcased as a robust framework, demonstrate their prowess in generating paths that not only optimize efficiency but also ensure safety in robotic navigation. Leveraging mathematical optimization algorithms, these methods engage in an iterative refinement of the trajectory. This iterative process takes into account a spectrum of factors. The ultimate result of these optimization-based approaches is the creation of a path that optimally navigates the conflicting objectives inherent in robotic navigation systems. This involves a delicate equilibrium between minimizing path length, ensuring safety through obstacle avoidance, and conserving energy. The synthesis of these considerations culminates in an enhanced overall performance of robotic navigation systems, marking a significant stride toward achieving seamless and effective autonomous navigation.

### 3.4. Time Cost in Robotic Navigation

Time cost in navigation encapsulates the duration it takes for a robot to execute a predetermined path within its operational constraints. The imperative to minimize this time cost emerges as a pivotal optimization criterion, intricately shaped by the dynamic interplay of the robot’s functionality and the environmental dynamics and kinematics. The overarching objective is to bolster navigation efficiency by curtailing the time required for precise path tracking.


**Time-Optimal Trajectory Planning (TOTP)**


At the forefront of time optimization strategies is **Time-Optimal Trajectory Planning (TOTP)**, a process finely tuned to craft a robot’s path for expeditious tracking of the predetermined trajectory. The pivotal juncture when the robot seamlessly adheres to the path while meeting operational requisites is termed the **execution time**. TOTP methodologies are strategically crafted to streamline path tracking, minimizing the time cost, which unfolds as a multifaceted function intricately woven with the robot’s dynamics and the environmental intricacies.

However, the resolution constraints inherent in both the robot and its environment pose formidable challenges to achieving TOTP, rendering modeling a formidable task. One pivotal approach involves maximizing velocity while judiciously considering constraints, mitigating the risk of undesirable jerking in the robot’s motion. The delineation of two primary time-optimal criteria, namely **continuous time** and **discrete time** problem definitions, further enriches the spectrum of time optimization strategies. Two types of time-optimal criteria can be defined for the general TOTP problem: continuous time and discrete time problem definitions. The **continuous time-optimal optimization problem** is defined as follows:(4)$$\min_{u(t)}\int_{0}^{T}1\;dt\;\mathrm{subject}\;\mathrm{to}\;\dot{x}\left(t\right)=f\left(x\left(t\right),u\left(t\right)\right),$$
where

f(x,u) is the state space of the robot dynamics;*x* is the state vector, consisting of position, velocity, and possibly acceleration;*u* is the control input that may depend on voltage; torque, or other functions of control manipulators;*T* is the total execution time.

The **discrete time-optimal optimization problem** is defined as follows:(5)$$\min_{u_{k}}\sum_{k=0}^{N}1\;\mathrm{subject}\;\mathrm{to}\;x_{k+1}=g\left(x_{k},u_{k}\right),$$
where

g(.) is the robot motion acceleration, which is a function of the control input $u_{k}$ at the *k*-th time sample;*N* is the final time step when the robot reaches the goal;$x_{k}$ is the *k*-th collision-free waypoint.

The methods for calculating the time cost of robotic navigation incorporate dynamic constraints, environmental factors, and optimization techniques to minimize the total execution time. Below are some key formulas used in recent research:


**Time Cost with Dynamic Constraints**


The time cost *J* for a trajectory can be calculated as:(6)$$J=\int_{0}^{T}\left(∥\dot{x}{\left(t\right)∥}^{2}+\lambda {∥u\left(t\right)∥}^{2}\right)dt,$$
where:$\dot{x}\left(t\right)$ is the velocity of the robot,u(t) is the control input,λ is a weighting factor that balances the trade-off between velocity and control effort.

This formulation ensures that the robot minimizes both the time and energy consumption during navigation [47]. The weighting factor λ in Equation (Equation 6) balances the trade-off between minimizing the robot’s velocity magnitude and the control effort, and its optimal value is typically selected through empirical tuning, multi-objective optimization, or domain-specific criteria to align with the navigation task’s priorities.


**Time Cost with Environmental Constraints**


In dynamic environments, the time cost must account for obstacles and environmental changes. A common approach is to use a penalty function:(7)$$J=\int_{0}^{T}\left(∥\dot{x}{\left(t\right)∥}^{2}+\lambda {∥u\left(t\right)∥}^{2}+\sum_{i}\phi_{i}\left(x\left(t\right)\right)\right)dt,$$
where

$\phi_{i}\left(x\left(t\right)\right)$ is a penalty function that increases the cost when the robot approaches obstacles or violates environmental constraints;λ is a weighting factor for the control effort.

This method ensures collision-free navigation while minimizing time cost [48].


**Time Cost in Multi-Robot Systems**


For multi-robot systems, the time cost can be extended to include coordination constraints:(8)$$J=\sum_{j=1}^{M}\int_{0}^{T}\left(∥{\dot{x}}_{j}{\left(t\right)∥}^{2}+\lambda {∥u_{j}\left(t\right)∥}^{2}+\sum_{k\neq j}\psi_{jk}\left(x_{j}\left(t\right),x_{k}\left(t\right)\right)\right)dt,$$
where

*M* is the number of robots;$\psi_{jk}\left(x_{j}\left(t\right),x_{k}\left(t\right)\right)$ is a penalty function that ensures collision avoidance between robots *j* and *k*.

This formulation is particularly useful in swarm robotics and collaborative tasks [49]. Recent research has introduced advanced methods to address the challenges of TOTP, including the following:


**Deep Reinforcement Learning (DRL)**


DRL-based approaches have been employed to learn time-optimal trajectories in complex environments [50]. These methods leverage neural networks to approximate the optimal policy, minimizing the time cost while satisfying dynamic constraints. The calculation criteria for DRL include the following:**Reward Function**: Designed to penalize time consumption and deviations from the desired trajectory.**State-Action Space**: Encodes the robot’s dynamics and environmental constraints.**Training Efficiency**: Measured by the convergence rate and computational resources required.


**Model Predictive Control (MPC)**


MPC frameworks have been extended to incorporate time-optimal constraints, enabling real-time trajectory optimization [26]. The calculation criteria for MPC include:**Horizon Length**: Determines the number of future steps considered in the optimization.**Constraint Handling**: Ensures the feasibility of the trajectory under dynamic and environmental constraints.**Computational Complexity**: Measured by the time required to solve the optimization problem at each time step.


**Multi-Objective Optimization**


Recent studies have combined time-optimality with other objectives, such as energy efficiency and safety, using Pareto optimization techniques [27]. The calculation criteria for multi-objective optimization include the following:**Pareto Front**: Represents the trade-off between competing objectives.**Weighting Factors**: Used to prioritize time-optimality over other objectives.**Scalability**: Evaluated based on the ability to handle high-dimensional state spaces.

The choice of the weighting factor λ is crucial, as it governs the trade-off between trajectory smoothness, energy efficiency, and responsiveness to dynamic changes. In practice, λ can be selected in several ways:**Empirical tuning:** Values of λ are adjusted experimentally until the robot achieves satisfactory performance in representative environments.**Optimization-based methods:** λ is treated as a hyperparameter and optimized through grid search, Bayesian optimization, or reinforcement learning to minimize a performance metric such as tracking error or energy usage.**Domain-specific constraints:** For safety-critical or resource-limited systems, λ may be constrained by hardware limits (e.g., maximum actuator torque) or mission requirements (e.g., prioritizing faster response over energy saving).

Thus, λ is not fixed universally but is determined by a balance between robot dynamics, environmental complexity, and task requirements.


**Adaptive TOTP**


Adaptive methods dynamically adjust the trajectory based on environmental changes, ensuring time-optimality in dynamic settings [51]. The calculation criteria for adaptive TOTP include the following:**Replanning Frequency**: Determines how often the trajectory is updated.**Convergence Speed**: Measures the time required to adapt to new environmental conditions.**Robustness**: Evaluated based on the ability to handle uncertainties and disturbances.


**Applications and Challenges**


Time-Optimal Trajectory Planning finds applications in various domains, including the following:Autonomous vehicles;Industrial robotics;Aerial drones.

However, challenges such as computational complexity, real-time adaptability, and the trade-off between time-optimality and other constraints (e.g., energy consumption) remain active areas of research.

In a nutshell, the aim of TOTP involves the consideration of crucial factors, including acceleration, velocity, jerk, and the dynamic characteristics of the robot. In defining the TOTP problem, two primary criteria are typically emphasized: continuous time and discrete time optimization. These criteria are essential for optimizing the performance of robotic systems, ensuring the shortest time for trajectory execution. Recent advancements, such as DRL, MPC, and adaptive methods, have further enhanced the capabilities of TOTP, paving the way for more efficient and robust robotic navigation systems.

### 3.5. Energy Cost in Robotic Navigation

The optimization of energy consumption in mobile robots is a multifaceted endeavor intricately tied to the dynamic and kinematic model of the robot and environmental characteristics. Formulating optimal control strategies necessitates a nuanced consideration of the interplay between these factors. A pivotal criterion for effective energy management involves the reduction of the generated path length. Additionally, environmental features play a crucial role in shaping an optimum governing strategy for energy consumption in mobile robots. Research indicates that minimizing unnecessary movements and deploying predictive control algorithms are key strategies for significant energy reduction. This holistic perspective encompasses the summation of various energy costs, including the following:**Kinetic Energy (Ek)**: Energy associated with the robot’s motion.**Traction Resistance Energy (Ef)**: Energy dissipated in overcoming traction resistances.**Motor Heating Energy (Ee)**: Energy lost as heat in the motors.**Mechanical Friction Energy (Em)**: Energy dissipated in overcoming friction torque.**Idle Energy (Eidle)**: Energy consumed by idling motors and onboard electric devices.

The total energy cost $E_{\mathrm{total}}$ of a mobile robot can be comprehensively defined as follows:(9)$$E_{\mathrm{total}}=\int_{0}^{T}P_{\mathrm{total}}\left(t\right)\;dt,$$
where

$P_{\mathrm{total}}\left(t\right)$ is the total power consumption and loss at time *t*;*T* is the total execution time.

This formulation provides a comprehensive metric for evaluating and optimizing the energy efficiency of mobile robots [30]. Recent research has introduced advanced methods to optimize energy consumption in robotic navigation, including the following:


**Predictive Energy Management**


Predictive control algorithms leverage environmental data to minimize energy consumption. The energy cost can be expressed as follows:(10)$$E_{\mathrm{predictive}}=\int_{0}^{T}\left(P_{\mathrm{motion}}\left(t\right)+P_{\mathrm{idle}}\left(t\right)\right)\;dt,$$
where

$P_{\mathrm{motion}}\left(t\right)$ is the power consumed during motion;$P_{\mathrm{idle}}\left(t\right)$ is the power consumed during idle states.

This approach has been shown to reduce energy consumption by up to 20% in dynamic environments [52].


**Regenerative Energy Recovery**


Regenerative braking systems recover kinetic energy during deceleration. The recovered energy $E_{\mathrm{regen}}$ can be calculated as:(11)$$E_{\mathrm{regen}}=\int_{0}^{T}\eta \cdot P_{\mathrm{braking}}\left(t\right)\;dt,$$
where:η is the efficiency of the regenerative braking system,$P_{\mathrm{braking}}\left(t\right)$ is the power generated during braking.

This method is particularly effective in electric and hybrid robotic systems [53].

### 3.6. Risk Cost in Robotic Navigation

Ensuring safety is a pivotal consideration in path planning algorithms, and it is essential to create a risk map that accurately assesses the level of danger associated with different routes. The development of such a risk map is indispensable for evaluating the potential hazards when traversing specific positions, taking into account the presence of obstacles. To construct a risk map, a grid of probabilities is established using two-dimensional coordinates. Within this framework, a probability of zero signifies a negligible collision risk, while a probability of one indicates a heightened likelihood of collision. The computation of risk costs is contingent upon the probability of unforeseen events, encompassing:**Collision Risk**: Probability of collisions with environmental elements or individuals.**Robot Malfunction**: Probability of robot failure or abrupt movements.**Environmental Hazards**: Probability of natural events such as rain or wind increasing the risk of slipping or crashing.

The risk level R(t) at a specific location r(t) can be calculated as:(12)$$R\left(t\right)=P_{\mathrm{event}}\left(t\right)\cdot C_{\mathrm{event}},$$
where:$P_{\mathrm{event}}\left(t\right)$ is the probability of a risk event at time *t*,$C_{\mathrm{event}}$ is the cost associated with the event.

The total risk cost $R_{\mathrm{total}}$ of a generated path from $r_{0}$ to $r_{\mathrm{goal}}$ can be expressed as:(13)$$R_{\mathrm{total}}=\int_{0}^{T}R\left(t\right)\;dt.$$

Recent studies have presented innovative techniques for evaluating risks in robotic navigation, which encompass:


**Neural Network-Based Risk Prediction**


Neural networks are used to predict collision probabilities based on historical data and real-time sensor inputs. The risk cost can be expressed as(14)$$R_{\mathrm{NN}}\left(t\right)=f_{\mathrm{NN}}\left(r\left(t\right),\theta \right),$$
where

$f_{\mathrm{NN}}$ is the neural network function;θ represents the network parameters.

This approach has been shown to improve risk prediction accuracy by up to 30% [54].


**Fuzzy Logic for Dynamic Risk Mapping**


Fuzzy logic systems are employed to handle uncertainties in dynamic environments. The risk cost can be calculated as follows:(15)$$R_{\mathrm{fuzzy}}\left(t\right)=\sum_{i}\mu_{i}\left(r\left(t\right)\right)\cdot C_{i},$$
where

$\mu_{i}\left(r\left(t\right)\right)$ is the membership function for the *i*-th risk factor;$C_{i}$ is the cost associated with the *i*-th risk factor.

This method is particularly effective in unstructured environments [55].

Figure 4 illustrates a concrete case example of a risk map within the context of path planning. This visual representation showcases the practical application of risk assessment and mapping techniques in determining optimal paths, providing valuable insights for navigating through dynamic environments safely and efficiently.

### 3.7. Integration of Optimization Criteria

The integration of these criteria into a unified optimization framework is a challenging yet essential task. Multi-objective optimization techniques are often employed to balance competing criteria, such as minimizing trajectory length while ensuring smoothness and collision avoidance [56]. Additionally, task-specific requirements, such as payload constraints or mission deadlines, may further influence the optimization process.

## 4. Approaches to Solving Optimal Navigation Problems

To address optimal navigation challenges, researchers use various methodologies, including numerical and analytical approaches. Analytical methods aim to minimize costs directly to obtain optimal solutions, but they can be challenging to apply in complex and constrained navigation scenarios. Therefore, researchers tend to use numerical solutions, which include sampling and sequence-based techniques such as greedy algorithms, Dynamic Programming, evolutionary algorithms, and sampling-based algorithms. These techniques have been applied in various studies, including multi-robot navigation, off-road navigation, and navigation under localization uncertainty [57].

### 4.1. Greedy Algorithms

Greedy algorithms are a type of optimization algorithm that selects the best immediate solution at each step without considering the entire problem. These algorithms are top-down approaches that select the locally optimal solution at each step without considering the previous step. They are simple to implement and understand, but do not guarantee the best overall solution and can be slow. Some classic sequence-based approaches of greedy algorithms include Depth First Search (DFS) [58], Dijkstra’s Algorithm [59], A* Algorithm [60], D* Algorithm [61], and Phi* Algorithm [62]. Greedy algorithms are commonly used for various applications, such as robotic navigation and problem-solving. While they can be ideal for problems with an optimal substructure, they may not always yield the best solution. The comparison provided in Figure 5 offers insights into the fundamental differences between the greedy algorithm and optimal approaches in navigation scenarios.

### 4.2. Dynamic Programming (DP)

Dynamic Programming (DP) strategically addresses optimal problems by breaking them down into more manageable sub-optimal problems, progressing towards the optimal solution step by step [57]. This process enhances efficiency and computational effectiveness. In the context of graph theory, DP formulations play a pivotal role in determining optimal sequences for visiting nodes within an adjacency graph associated with a decomposed environment. Figure 6 provides a general overview of dynamic programming in the context of navigation.

### 4.3. Evolutionary Algorithms (EAs)

Evolutionary algorithms (EAs) randomly select a candidate set of solutions and apply the quality function as an abstract fitness measure. EAs utilize a fitness function to find the optimal solution by converging from the initial state to the global optimal. These algorithms, including Particle Swarm Optimization (PSO) [63], Ant Colony Optimization (ACO) [64], and Genetic Algorithm (GA) [65], fall under the umbrella of EAs. Particularly effective in addressing multi-objective problems, they offer viable solutions for optimizing path planning scenarios with diverse criteria. The optimization process involves iteratively converging from an initial state towards the global optimum using a fitness function. However, it is essential to note that the inclusion of constraints can introduce computational challenges for these algorithms. Constraints are factors that restrict the feasible solution space, potentially increasing the computational burden. Despite this, EAs remain powerful tools for solving complex optimization problems, providing valuable solutions in various domains. Recent state-of-the-art (SOTA) approaches combine EAs with hybrid strategies, multi-objective optimization, and adaptive mechanisms to improve path quality, efficiency, and robustness, as summarized in Table 2.

### 4.4. Sampling-Based Algorithms

Sampling-based algorithms compromise between greedy algorithms and exploring data in unknown and large areas, making them suitable for high-dimensional and complex problems with low computational costs. Probabilistic Roadmap (PRM) [75], Rapidly-exploring Random Tree (RRT) [76], and Rapidly-exploring Random Tree Star (RRT*) [77] are some of the popular sampling-based algorithms. However, these algorithms may fail in dynamic environments. Recent advancements, such as Adaptive RRT* [78] and Deep Sampling-Based Planning [79], have improved their performance in dynamic and uncertain environments. Table 3 summarizes recent developments in sampling-based navigation algorithms.

By and large, sampling-based motion planners offer a unique balance of strengths and limitations compared to other planning methods.


**Key Advantages:** These planners are scalable and flexible, making them suitable for high-dimensional spaces (e.g., robotic manipulators) where grid-based searches are computationally prohibitive. They do not require explicit environment discretization, enabling efficient path generation in complex, cluttered settings. Modern extensions, such as Adaptive RRT* and deep learning-guided planners, further enhance adaptability by dynamically adjusting sampling strategies in response to moving obstacles or semantic environmental cues.**Limitations:** Despite their strengths, standard sampling-based methods can produce sub-optimal, non-smooth paths, often requiring post-processing. They also struggle in dynamic environments without specific adaptations and are sensitive to parameter tuning (e.g., step size, sampling distributions). Additionally, learning-based enhancements, while improving adaptability, can introduce extra computational overhead from model training and real-time inference, potentially affecting performance.


### 4.5. Recent Advances in Optimal Navigation

Recent research has introduced advanced methods to address the challenges of optimal navigation, including the following:**Deep Reinforcement Learning (DRL)**: DRL-based approaches have been employed to learn optimal navigation policies in complex environments [85].**Model Predictive Control (MPC)**: MPC frameworks have been extended to incorporate dynamic constraints, enabling real-time navigation optimization [86].**Multi-Objective Optimization**: Recent studies have combined navigation objectives, such as time-optimality and energy efficiency, using Pareto optimization techniques [87].**Adaptive Navigation**: Adaptive methods dynamically adjust navigation strategies based on environmental changes, ensuring robustness in dynamic settings [88].

The selection of path planning strategies and styles should consider the characteristics of the robot and environments, as well as priorities and tasks. A general comparison of different path planning architectures is provided by [89]. Recent advancements in greedy algorithms, dynamic programming, evolutionary algorithms, and sampling-based algorithms have significantly improved the efficiency and robustness of optimal navigation systems. These approaches, combined with modern techniques such as DRL and MPC, pave the way for more effective and adaptive navigation solutions in complex and dynamic environments.

## 5. Overview of Collision-Free Path Planning Strategy

To craft robust path planning algorithms, a meticulous understanding of both priorities and constraints specific to the problem is essential. Beyond prioritization, the path planning algorithm design must address various constraints inherent in the system. The process involves a comprehensive evaluation of factors influencing the navigation system’s performance. Figure 7 provides valuable insight into the overarching block diagram of a navigation system, serving as a guide to identify additional elements crucial for an inclusive design approach. This holistic perspective ensures that the developed algorithms align with the intricate requirements of the navigation environment, fostering efficiency and adaptability.

Assuming the robot is equipped with a tracking controller and relies on monitored motion, global map data, real-time sensory information, and obstacle positions, Figure 7 emphasizes the generation of a risk map matrix and the design of a path planning algorithm. While optimizing path length is a priority, the limitations of global approaches in ensuring safety within dynamic environments are acknowledged. To address this, a hybrid path planning approach integrates reactive and classic methods. This combination offers a robust solution to navigate challenges in dynamic environments, where real-time adaptation is crucial for ensuring both efficiency and safety. This approach can benefit from the advantages of both classic and reactive methods while reducing their deficiencies.

### 5.1. Hybrid Path Planning

Hybrid path planning combines the strengths of global and local planning methods to achieve robust navigation in dynamic environments. A typical hybrid approach can be formulated as follows:(16)$$P_{\mathrm{hybrid}}=\alpha \cdot P_{\mathrm{global}}+\left(1-\alpha \right)\cdot P_{\mathrm{local}},$$
where

$P_{\mathrm{hybrid}}$ is the hybrid path;$P_{\mathrm{global}}$ is the globally optimal path;$P_{\mathrm{local}}$ is the locally adjusted path;α is a weighting factor that balances global and local planning.

In hybrid path planning, the weighting factor α plays a pivotal role in determining how much influence the globally optimal path $P_{\mathrm{global}}$ has compared to the locally adjusted path $P_{\mathrm{local}}$. Rather than keeping α fixed, recent research emphasizes the importance of adaptively tuning this factor in real-time, allowing robots to balance long-term optimality with short-term responsiveness under dynamic and uncertain conditions.

Several adaptive strategies have been proposed in the literature:**Behavior-based methods:** These approaches adjust α depending on environmental cues such as obstacle density or the need for smoother local maneuvers. For instance, heuristic and bio-inspired algorithms like the Hybrid Improved Artificial Fish Swarm Algorithm (HIAFSA) dynamically regulate α based on proximity to obstacles [90].**Optimization-driven methods:** Here, α is updated online through explicit optimization frameworks. A notable example is the adaptive visibility graph combined with A*, which minimizes tracking error and energy consumption by recalibrating α in response to evolving constraints [91].**Learning-based approaches:** More recent methods rely on reinforcement learning and evolutionary strategies to “learn” how α should evolve across diverse scenarios. The Multi-strategy Hybrid Adaptive Dung Beetle Optimization (MSHADO) algorithm, for example, employs chaotic mapping and multi-strategy fusion to enhance population diversity, enabling robots to adaptively balance global exploration with local exploitation [92].

These adaptive mechanisms highlight that α is not merely a tuning constant, but a dynamic variable central to achieving both robust global guidance and agile local responsiveness in hybrid path planning frameworks.

### 5.2. Real-Time Adaptation

Real-time adaptation is crucial for ensuring collision-free navigation in dynamic environments. One approach involves the use of Model Predictive Control (MPC) to continuously update the robot’s trajectory based on real-time sensory data. The MPC formulation can be expressed as follows:(17)$$\min_{u(t)}\sum_{t=0}^{T}∥x\left(t\right)-x_{\mathrm{ref}}{\left(t\right)∥}^{2}+{∥u\left(t\right)∥}^{2},$$
where

x(t) is the robot’s state at time *t*;$x_{\mathrm{ref}}\left(t\right)$ is the reference trajectory;u(t) is the control input;*T* is the prediction horizon.

The specific form of u(t) depends on the system dynamics and the navigation problem under consideration. For instance, in a unicycle model(18)$$u\left(t\right)=\left[\begin{array}{c} v(t)\\ \omega (t) \end{array}\right],$$
where v(t) is the linear velocity and ω(t) is the angular velocity. In a differential-drive robot, u(t) may correspond to the left and right wheel velocities, while for holonomic robots, it can include translational components along multiple axes. In motion control formulations, u(t) may directly represent the applied torques, whereas in embedded tracking controllers, it can denote the next desired position of the robot at each prediction step.

The inclusion of the control penalty ${∥u\left(t\right)∥}^{2}$ in the cost function ensures smooth and feasible motion by discouraging aggressive actuation, while the tracking term $∥x\left(t\right)-x_{\mathrm{ref}}{\left(t\right)∥}^{2}$ enforces accurate trajectory following. Thus, u(t) is central to balancing tracking accuracy, energy efficiency, and dynamic feasibility in the MPC framework.

Recent advancements in MPC include the integration of deep learning models to improve prediction accuracy [93].

### 5.3. Recent Advancements in Collision-Free Path Planning

Recent research has introduced advanced methods to address the challenges of collision-free path planning, including the following:**Deep Reinforcement Learning (DRL)**: DRL-based approaches have been employed to learn collision-free navigation policies in complex environments [94].**Multi-Agent Path Planning**: Techniques for coordinating multiple robots to avoid collisions while achieving individual goals [95].**Uncertainty-Aware Planning**: Methods that account for uncertainties in sensor data and environmental dynamics [84].**Energy-Efficient Path Planning**: Energy-efficient path planning combines the optimization of energy consumption with collision avoidance, employing various algorithmic strategies tailored to different robotic systems and environments. These approaches address challenges like terrain roughness, multi-agent coordination, and dynamic obstacles while minimizing motion costs [82].

The design of collision-free path planning algorithms requires a holistic approach that considers both global and local constraints. Recent advancements in risk map generation, hybrid path planning, and real-time adaptation have significantly improved the robustness and efficiency of navigation systems. These approaches, combined with modern techniques such as DRL and MPC, pave the way for more effective and adaptive collision-free navigation solutions in complex and dynamic environments.

## 6. Challenges in Geometric Optimal Navigation

Geometric optimal navigation involves finding the shortest or most efficient path for a robot while avoiding obstacles and adhering to dynamic constraints. However, several challenges arise in this domain, particularly in high-dimensional spaces, real-time computation, and the integration of perception systems. This section discusses these challenges, recent advancements, and their implications for robotic navigation.

### 6.1. Scalability in High-Dimensional Spaces

High-dimensional spaces pose significant computational challenges for path planning. As the dimensionality of the environment increases, the computational complexity grows exponentially, making it difficult to find optimal paths efficiently. Recent studies have explored dimensionality reduction techniques and machine learning to address this issue [83].

#### 6.1.1. Dimensionality Reduction Techniques

Dimensionality reduction techniques, such as Principal Component Analysis (PCA) and t-Distributed Stochastic Neighbor Embedding (t-SNE), have been applied to reduce the complexity of high-dimensional spaces. Mathematically, PCA can be expressed as(19)Y=X·W,
where

X is the high-dimensional data matrix;W is the transformation matrix;Y is the low-dimensional representation.

These techniques help reduce the computational burden while preserving the essential structure of the environment [80].

#### 6.1.2. Machine Learning Approaches

Machine learning approaches, such as deep autoencoders, have been used to learn compact representations of high-dimensional spaces. The autoencoder loss function is given by(20)$$L=∥X-\hat{X}{∥}^{2},$$
where $\hat{X}$ is the reconstructed data. These methods enable efficient path planning in high-dimensional environments by learning low-dimensional embeddings [81].

### 6.2. Real-Time Computation for Dynamic Environments

Real-time navigation in dynamic environments involves solving optimal path planning problems under stringent timing requirements while accounting for moving obstacles, robot dynamics, and environmental uncertainty. As problem complexity scales with environment size, obstacle density, and number of agents, scalable algorithmic techniques become vital [96]. The *optimal navigation problem* in a dynamic environment can be formulated as follows:(21)$$\begin{array}{cc} \min_{x(t),u(t)}\; & J=\int_{0}^{T}c\left(x(t),u(t),t\right)\;dt\\ s.t.\; & \left\{\begin{array}{cc} \dot{x}\left(t\right)=f\left(x\left(t\right),u\left(t\right),t\right) & \left(\mathrm{dynamics}\right)\\ x(t)\notin O(t) & (\mathrm{collision}\;\mathrm{avoidance})\\ x\left(0\right)=x_{\mathrm{start}},\;x\left(T\right)=x_{\mathrm{goal}} & (\mathrm{boundary}\;\mathrm{conditions})\\ u(t)\in U & (\mathrm{control}\;\mathrm{limits}) \end{array}\right. \end{array}$$
where x(t) is the robot state trajectory, u(t) are controls, c(·) is a cost integrating path length, smoothness, energy, or safety margins, and O(t) models time-varying obstacles.

Scalability challenges arise from the curse of dimensionality, dynamic constraints, and frequent replanning triggered by environmental changes.

**Overall scalability strategies include the following**:Parallelizing computationally intensive steps (sampling, collision checking, graph search).Adaptive spatial and temporal resolution to reduce unnecessary computation.Decomposing the problem into hierarchical or modular sub-problems.Leveraging learned models to prune search space or estimate costs rapidly.

### 6.3. Parallel Optimization for Real-Time Trajectory Planning

Optimization-based planners seek continuous trajectories by minimizing a cost function under dynamic and environmental constraints. While effective in theory, traditional solvers often converge too slowly for real-time navigation, limiting their use in dynamic environments.

Parallel gradient-based optimization addresses these challenges by leveraging multi-core CPUs or GPUs to accelerate cost and gradient evaluations. These methods enable planners to meet strict real-time requirements while maintaining trajectory quality, as described in Algorithm 1.
**Algorithm 1** Real-Time Parallel Trajectory Optimization  1:**Input:** Previous trajectory x(t), control inputs u(t) (warm-start)  2:**while** within time budget and convergence not met **do**  3:    **Parallel:** For each knot along the trajectory:  4:       Compute gradients for smoothness, obstacle cost, and dynamics  5:       Compute constraint gradients (e.g., collision or control limits)  6:   Aggregate gradients  7:   Update x(t) and u(t) (e.g., via projected gradient descent)  8:   Project updated trajectory to enforce hard constraints  9:**end while**10:**Output:** Refined trajectory x*, controls u*

### 6.4. Strategies for Real-Time Scalability

Effective real-time trajectory optimization relies on a combination of algorithmic innovations and hardware acceleration. The following strategies form the backbone of scalable real-time planners:**GPU-Accelerated Evaluation:** Modern GPUs allow hundreds to thousands of trajectory knot gradients to be evaluated simultaneously, enabling speed-ups by orders of magnitude compared to serial CPUs. This parallelism dramatically reduces iteration times for gradient computation and constraint checking, empowering planners to meet real-time deadlines even in high-dimensional state spaces [97,98,99].Recent work by Rastgar [98] proposes novel GPU-parallel optimization algorithms that adapt constraint formulations to fully leverage GPU architectures, markedly enhancing scalability and robustness in dynamic scenarios. Similarly, Yu et al. [99] introduce TOP, trajectory optimization via parallel consensus ADMM that achieves near-constant time complexity per iteration by decomposing long trajectories into parallelizable segments, enabling large-scale real-time path planning on GPUs.**Warm Starting:** Utilizing the previously computed optimal trajectory as an initial guess accelerates the convergence of iterative solvers. This approach is especially effective in dynamic environments where consecutive plans differ only slightly [98,100].By reusing prior solutions, planners reduce redundant computation while improving continuity and smoothness of resulting trajectories. This practical technique aligns with state-of-the-art GPU-accelerated approaches that integrate warm-starting for real-time feasibility.**Receding Horizon Execution:** Instead of optimizing over the entire trajectory horizon, planners focus on a shorter, fixed-duration segment. Only the initial portion of the plan is executed before replanning occurs, maintaining continual responsiveness to environmental changes and dynamic obstacles [98,101].This limited horizon approach bounds computational demands, facilitates faster replanning, and supports adaptive trajectory refinement, key for scalable navigation in cluttered or rapidly changing environments.**Adaptive Termination and Step Sizing:** Optimization algorithms dynamically adjust their step sizes and employ early stopping criteria once trajectories reach a suitable quality level within the operational time budget. This balances the trade-off between latency and solution optimality.For example, algorithms may terminate as soon as collision-free smooth paths are found, even if not perfectly optimal, ensuring timely availability of actionable plans without compromising safety [99].**Integrated Recent Advances:**Incorporating the above core principles, recent methods extend scalability and robustness in real-time planning:–**Semantic-Aware Optimization:** He et al. [102] develop a spatio-temporal semantic graph optimizer tailored for urban autonomous driving. Their approach handles dynamic obstacle semantics through sparse graph formulations, enabling real-time feasible trajectories that intelligently incorporate semantic understanding of traffic participants and road elements.–**Hybrid Sampling and Rewiring:** Silveira et al. [103] propose RT-FMT, a hybrid of Fast Marching Tree and RT-RRT*, that combines incremental rewiring and local-to-global tree reuse for faster execution and improved path quality in dynamic environments. By efficiently reusing prior tree structures and limiting expansion scope, RT-FMT exemplifies algorithmic decomposition, enabling scalability.–**Piecewise Parallel Optimization:** Yu et al. [99] introduce the TOP framework, which decomposes long trajectories into smaller segments solved in parallel while ensuring high-order continuity through consensus constraints. Deploying this method on GPUs supports extremely large-scale and long-horizon real-time trajectory optimization, pushing the frontier of computational performance in robotics.–**Constraint Reformulation for Parallelism:** Rastgar’s thesis [98] innovates by remodeling kinematic and collision constraints to be more amenable to parallel GPU computation, thereby enhancing planner scalability and resulting in more reliable real-time performance. GPU acceleration leverages parallel processing in graphics units to speed up computation-heavy tasks like sensor data processing and path planning, enabling real-time navigation in complex environments. Neural approximators, such as neural networks running on GPUs or neuromorphic chips, offer efficient models that reduce computation while preserving accuracy. These technologies are crucial for power-limited robots and edge devices, allowing them to perform advanced navigation tasks with manageable latency and energy usage [104,105,106].

Together, these innovations illustrate a robust framework for trajectory optimization under real-time constraints, combining modern parallel hardware, algorithmic acceleration, and semantic-aware planning to maintain low latency and high fidelity in dynamic, obstacle-rich environments.

### 6.5. Integration of Perception Systems

Accurate environment modeling relies on the seamless integration of perception systems, such as LiDAR, cameras, and IMUs. Recent advancements in multi-sensor fusion and deep learning have improved perception accuracy [107].

#### Deep Learning for Perception

Deep learning models, such as convolutional neural networks (CNNs), have been used to process sensory data and extract meaningful features. The loss function for training a CNN is given by(22)$$L=\sum_{i=1}^{N}{∥y_{i}-{\hat{y}}_{i}∥}^{2},$$
where $y_{i}$ and ${\hat{y}}_{i}$ are the ground truth and predicted values, respectively. These models improve perception accuracy in complex environments [108].

Geometric optimal navigation is a critical component of robotic systems, enabling efficient and safe movement in complex environments. However, future research should focus on developing more adaptive and energy-efficient navigation strategies, as well as improving the robustness of perception systems in uncertain and dynamic environments. By addressing these challenges, we can pave the way for more effective and reliable robotic navigation systems.

### 6.6. Ethical and Safety Concerns in Human–Robot Interactions

Ethical and safety concerns are critical for the real-world deployment of autonomous systems. Recent research has focused on developing ethical frameworks and safety protocols for human–robot interactions. Ethical frameworks guide robots to act in ways that respect societal norms and human rights. Core principles include the following:**Transparency**: Robots should clearly communicate intentions and behaviors to enhance human trust.**Accountability**: Developers and operators must be accountable for system behavior, particularly in high-stakes contexts.**Fairness**: Systems should actively mitigate bias and promote equitable treatment across user demographics.

Ethically-aligned systems must reconcile trade-offs between efficiency, safety, and moral responsibility [109,110]. Recent research emphasizes formalizing these principles in algorithms for real-time navigation and decision-making [103].

Beyond ethical reasoning, physical safety remains a critical design constraint. Key safety mechanisms include the following:**Collision Avoidance**: Sensor fusion and predictive models allow robots to anticipate and avoid human contact.**Emergency Stop Mechanisms**: Systems must be capable of halting immediately under risk conditions.**Human-in-the-Loop Control**: Dynamic shared autonomy enables humans to intervene in uncertain or dangerous scenarios.

Recent developments use machine learning to dynamically assess and adapt to human proximity and behavior, improving safety in unpredictable settings [111,112,113].

To translate abstract ethical principles into practical algorithm designs, we propose a clear, step-wise framework that integrates ethics and safety throughout the path planning process [114,115,116,117]. The framework begins by identifying stakeholders and eliciting their safety and equity requirements, grounding the system in real social needs. Measurable metrics such as probabilistic risk scores, fairness constraints, and safety limits are then formalized to guide planning decisions.

Uncertainty and risk are modeled using stochastic predictions of dynamic agents, enabling the computation of per-trajectory risk assessments that inform ethical trade-offs. Ethics are embedded via multi-objective cost augmentation, supervisory filters to reject ethically unacceptable trajectories, or policy learning methods that internalize fairness and safety goals. Real-time feasibility and explainability are ensured by optimizing computational efficiency and producing human-interpretable justifications for decisions. Validation through diverse scenario testing and continuous audit ensures robustness and accountability.

A concrete example is the ethical trajectory planning algorithm by Geißlinger et al. [114] which encodes five ethical principles—risk minimization, priority for the worst-off, equal treatment, responsibility, and acceptable risk thresholds—into the trajectory evaluation. Through probabilistic risk modeling, the planner scores multiple candidate paths, penalizing those that unfairly endanger vulnerable users or exceed risk limits. This results in fair, safe trajectories selected transparently in real-time, validated across thousands of scenarios.

### 6.7. Toward Human-Centric Design and Learning

Human-centric design integrates physical, cognitive, and emotional dimensions of safety:**Safe Reinforcement Learning (Safe RL)**: Safety-aware RL ensures that robots learn policies without violating safety constraints—even in exploration phases [111,118].**Context-Aware Adaptation**: Integrating human feedback and context improves trust and user comfort in collaborative robotics [113,119].

Future systems must adopt unified frameworks that combine ethical reasoning, real-time risk assessment, and safe learning under uncertainty [102,103].

While progress is promising, significant challenges remain:**Integration Complexity**: Embedding ethical and safety modules into fast, real-time planners on resource-constrained platforms remains difficult.**Standardization Gaps**: Harmonization of international safety and ethics standards—such as ISO 12100 [120], which outlines general principles for machinery risk assessment and reduction, and ISO/TS 15066 [121], which defines collaborative robot safety requirements—with learning-based models is still ongoing [112]. These standards serve as global benchmarks for ensuring that autonomous and robotic systems meet essential safety and risk-reduction criteria while interacting with humans.**Human Perception**: Models often neglect psychological dimensions such as perceived safety and emotional response.

Looking forward, future research should aim to integrate ethical decision-making algorithms, robust safe reinforcement learning methods, and human-centric design principles into cohesive frameworks. These frameworks must balance operational efficiency with respect for human values and dynamic risk management, paving the way for autonomous robotic systems that are reliable, trustworthy, and ethically aligned [102,103,110].

By addressing these multifaceted challenges, the robotics community can accelerate the deployment of effective, reliable, and ethically sound robotic navigation and collaboration systems, fostering safer and more harmonious human–robot coexistence.

## 7. Enhancing Geometric Path Planning Through Optimization and Machine Learning

Optimization and machine learning (ML) techniques significantly enhance geometric path planning by combining precise trajectory computation with adaptive decision-making. Multi-modal Model Predictive Control (MMPC) combined with Q-Learning, as in the MAR-MPC framework, improves feasibility, enlarges the convergence region, and ensures collision-free navigation [26]. Reinforcement learning approaches, such as Deep Deterministic Policy Gradient (DDPG), allow robots to adaptively handle dynamic uncertainties, improving path smoothness and convergence [27].

Advanced methods further integrate optimization with learning-based solvers. The PTP RSNN-based predictive neuro-navigator combines log-convex optimization with kinematic and collision constraints, often using multi-objective formulations, constrained optimization, or Lagrangian methods to achieve optimal or near-optimal solutions [27]. Similarly, the smooth PSO-IPF navigator uses Particle Swarm Optimization with inverse predictive filtering under kinematic constraints to generate smooth and feasible trajectories [63]. Hybrid strategies that integrate ML-based real-time obstacle avoidance with geometric planners such as S-RRT [122] or GA-ACO [123] achieve robust, efficient, and safe path planning in dynamic environments.

Overall, this synergy highlights a core principle: optimization ensures precise geometric trajectories, while learning and adaptive solvers expand the feasible region, improve robustness, and enable mobile robots to navigate efficiently and reliably in complex, dynamic environments.

## 8. Conclusions

This survey presents a comprehensive synthesis of geometric optimal navigation and path planning strategies, with an emphasis on scalability, real-time performance, and adaptability to dynamic environments. By unifying classical geometric methods with contemporary optimization techniques and learning-based models, including GPU-accelerated frameworks and neural approximators, this paper underscores the significant strides made toward real-time, scalable autonomous navigation.

The reviewed literature reveals a clear trend toward hybrid approaches—combining hierarchical decomposition, parallel computation, and learning-augmented modules—to tackle increasing computational demands and environmental complexity. Recent advancements, such as attention-based neural networks, particularly Transformer architectures, offer new pathways to represent spatial–temporal dependencies and plan in high-dimensional, uncertain spaces more efficiently.

This work also highlights the importance of ethical and safety considerations in deploying optimization-based navigation systems. The real-world deployment of autonomous agents in safety-critical domains (e.g., healthcare robotics, autonomous vehicles, drones) necessitates guarantees on bounded sub-optimality, resilience to adversarial conditions, and alignment with human-centric values.

### Key Takeaways for Future Research

**Transformer-Based Planning:** Future systems should explore integrating spatial–temporal attention mechanisms into planning pipelines to improve generalization and context-aware navigation, especially in multi-agent and partially observable environments.**Real-Time and Embedded Efficiency:** Further innovation is needed to support GPU and neuromorphic execution on power-constrained platforms, ensuring autonomy is feasible for small-scale robots and edge devices.**Scalable Multi-Agent Coordination:** Scalability remains a bottleneck. Approaches that combine decentralized optimization, learning-based approximations, and adaptive communication protocols are promising.**Safe and Ethical Optimization:** New planning frameworks should incorporate constraints and verification layers that explicitly account for safety, fairness, and human preferences, particularly when operating alongside humans.**Standardization and Benchmarking:** To assess progress meaningfully, standardized evaluation frameworks and real-world benchmarks—particularly those involving uncertainty, real-time constraints, and ethical dilemmas—must be developed.

In summary, the convergence of parallel computing, geometric control theory, and deep learning promises robust, real-time path planning in dynamic environments. However, ensuring ethical behavior, safety guarantees, and hardware efficiency will remain central challenges in realizing truly autonomous, intelligent agents.

## Figures and Tables

**Figure 1 sensors-25-06874-f001:**
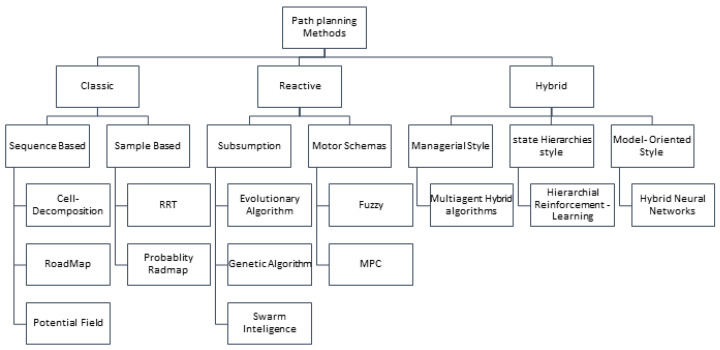
Integration of classical and reactive structures in hybrid path planning methods.

**Figure 2 sensors-25-06874-f002:**
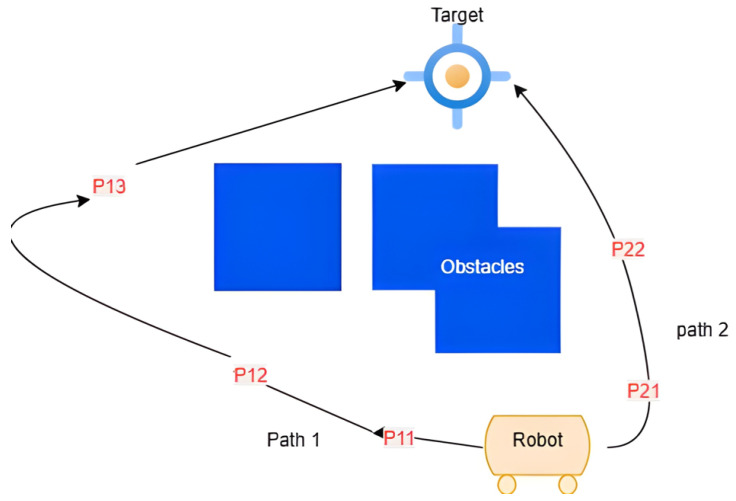
Two possible paths from an initial state to a target destination. Path 1 consists of 3 intermediate points, while Path 2 consists of 2 intermediate points.

**Figure 3 sensors-25-06874-f003:**
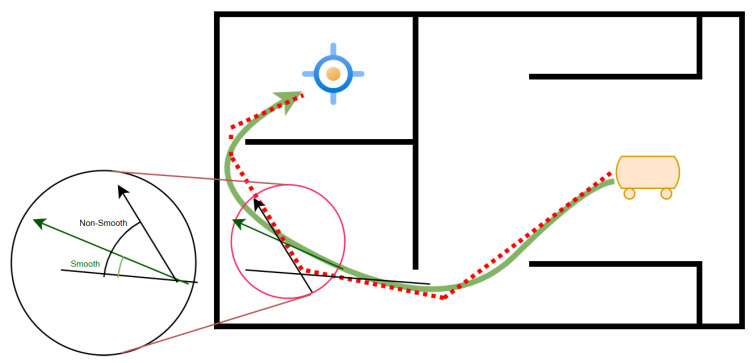
Movement angle difference in smooth path and non-smooth path. The green line indicates the smooth path and the red dash-line shows the non-smooth path.

**Figure 4 sensors-25-06874-f004:**
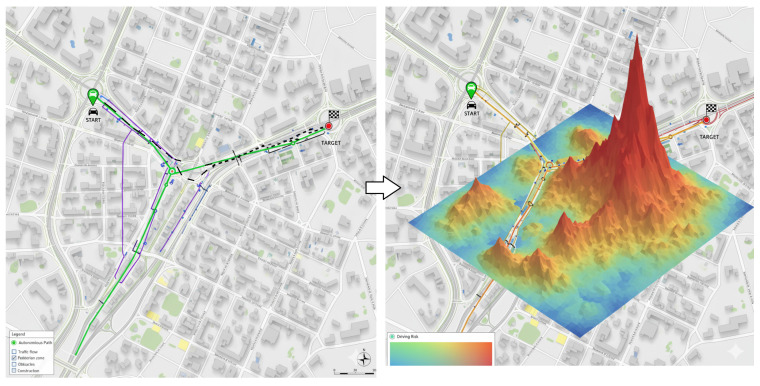
Risk map—Example of risk-aware navigation formulation. The left panel illustrates multiple candidate navigation paths from the Start to the Target, with different colored lines representing alternative routes generated by the path-planning algorithm (green: optimal path, purple: secondary paths, and blue: exploratory paths). The legend indicates that the urban map includes pedestrian zones, intersections, and road lanes that were considered in the navigation process. The right panel presents the corresponding risk map, where the height and color intensity represent the computed pedestrian–vehicle collision risk and other environmental obstacles along the route (from low—blue—to high—red). The optimal route is selected by minimizing the integrated pedestrian and obstacle collision risk across the path.

**Figure 5 sensors-25-06874-f005:**
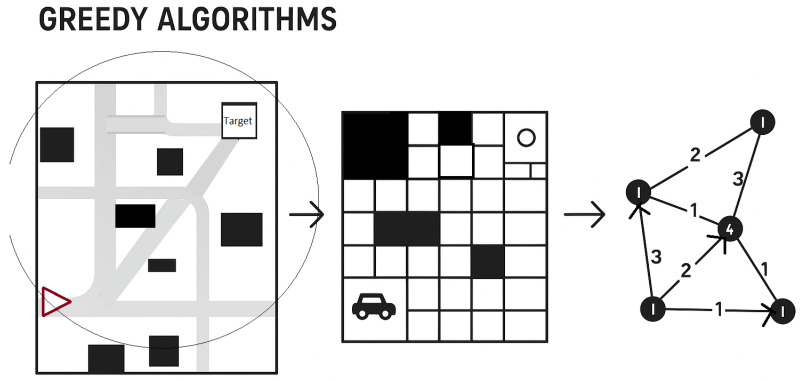
Greedy algorithm pipeline: (**Left**) The real-world environment is represented as a map with obstacles and a target location. (**Center**) The environment is discretized into a grid map, with cells marking obstacles and accessible spaces. (**Right**) The grid is converted into a graph, where numbered circles (1–4) correspond to sampled key nodes (locations in the environment), and edge labels indicate transition costs used by the greedy algorithm to select the next step toward the target. This progression demonstrates how spatial navigation problems are systematically transformed for algorithmic decision-making and cost evaluation.

**Figure 6 sensors-25-06874-f006:**
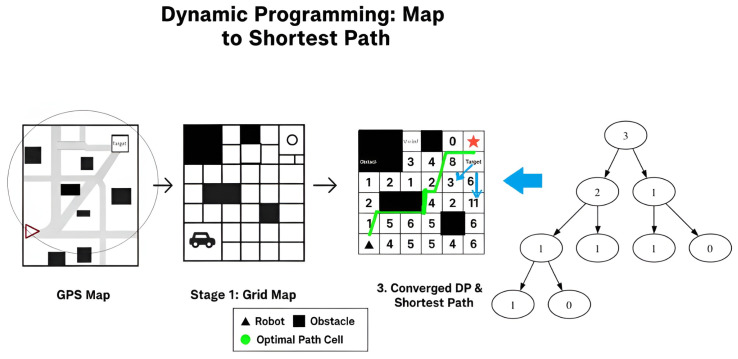
Dynamic programming pipeline for shortest path planning: (**Left**) The GPS map shows the robot’s position, obstacles, and the target. (**Center**) The environment is discretized into a grid, with black cells representing obstacles. (**Right**) Dynamic programming iteratively calculates the minimum cost-to-go (numbers 1–6) from each cell to the target (star) in a reverse manner—starting from the goal and expanding outward—and marks the optimal path cells used to reach the target. The blue arrow illustrates this backward calculation mechanism from the target to the start. (Far right) The solution is mapped onto a decision tree for policy extraction or further analysis. Symbols: black triangle = robot, black square = obstacle, green = optimal path cell, star = target.

**Figure 7 sensors-25-06874-f007:**
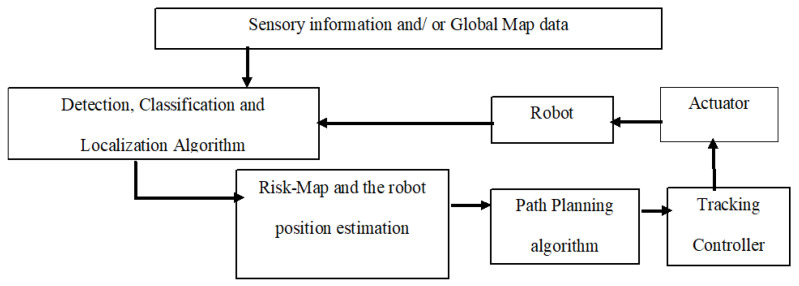
Overview of a collision-free path planning system diagram.

**Table 1 sensors-25-06874-t001:** Well-known path smoothing methods with practical considerations.

Method	Mathematical Definition	Parameters and Description	Computational Complexity	Applicability & Limitations
**Bézier Curve** [32,34]	B(t)=∑i=0nni(1−t)n−itiPi	$t\in \left[0,1\right],P_{i}$ (control points), *n* (degree). Smooth curve defined by control points.	Low	Easy to implement; good for offline planning; limited flexibility for complex dynamic obstacles.
**Elastic Stretching** [46]	${\min\int ∥\kappa \left(s\right)∥}^{2}ds$	κ(s) (curvature), *s* (arc length). Minimizes total squared curvature.	Medium	Produces very smooth paths; may be computationally heavy for long paths; sensitive to obstacle density.
**Minimum Angle Difference** [32]	$\min∥\theta_{i+1}-\theta_{i}∥$	$\theta_{i}$ (angle at waypoint *i*). Minimizes angular difference.	Low	Simple and fast; may not fully smooth paths in complex environments.
**Curvature Continuity** [42]	$\min\int ∥\kappa^{\prime }{\left(s\right)∥}^{2}ds$	$\kappa^{\prime }\left(s\right)$ (derivative of curvature). Ensures continuous curvature.	Medium–High	Smooth trajectories with continuity; requires more computation; may struggle with dense obstacles.
**Jerk Minimization** [43]	${\min\int ∥j\left(t\right)∥}^{2}dt$	j(t) (jerk at time *t*). Minimizes jerk.	Medium	Improves robot stability and comfort; may be computationally intensive in real-time control.
**Energy-Efficient Smoothing** [44]	${\min\int ∥F\left(v\left(t\right)\right)∥}^{2}dt$	F(v(t)) (force or energy as a function of velocity). Optimizes energy usage.	Medium–High	Useful for battery-constrained robots; trade-off with path length or time efficiency.
**Adaptive Smoothing** [45]	${\min\int ∥\kappa \left(s,t\right)∥}^{2}ds$	κ(s,t) (curvature at arc length *s* and time *t*). Adjusts smoothness dynamically.	High	Suitable for dynamic environments; computationally demanding; requires real-time updates.

**Table 2 sensors-25-06874-t002:** SOTA approaches in evolutionary algorithms for navigation and path planning.

Approach/Algorithm	Application Area	Main Advantages/Findings	Citation
Twin-Reinforced Chimp Optimization + Evolutionary Programming	Robot path planning	Outperforms other meta-heuristics in path length, consistency, time complexity, and success rate	[66]
Improved PSO with Evolutionary Operators (IPSO-EOPs)	Multi-robot navigation	Superior to DE and standard PSO in arrival time, safety, and energy use	[67]
Many-Objective EAs (HypE, GrEA, KnEA, NSGA-III)	Agricultural robot route planning	HypE delivers best performance for minimizing navigation cost and turning angle	[68]
Decomposition-based Multi-Objective EA (M2M-DW)	UAV path planning	Effectively handles constraints and infeasible solutions, reliable in complex scenarios	[69,70]
Multi-Objective Evolutionary PSO (MOEPSO)	Mobile robot path planning	Finds shortest, smoothest, and safest paths in static and dynamic environments	[71]
Bi-level Co-evolutionary Genetic Algorithm (IGA-CPP)	Coverage path planning	Efficient for irregular regions, fast convergence, optimized path length	[72]
NSGA-II and Multi-Objective EAs	Mobile robot navigation	NSGA-II excels in balancing path time and smoothness across diverse environments	[73]
Distributed Multi-Population EA	Maritime navigation	Multi-population approach improves solution quality over single-population EAs	[74]

**Table 3 sensors-25-06874-t003:** Recent sampling-based navigation algorithms.

Method	Formula	Parameter Definition
**Adaptive RRT*** [80]	$x_{\mathrm{new}}=x_{\mathrm{near}}+\eta \frac{x_{\mathrm{rand}}-x_{\mathrm{near}}}{∥x_{\mathrm{rand}}-x_{\mathrm{near}}∥}$ $\eta_{k+1}=\alpha \eta_{k}+\left(1-\alpha \right)f\left(\mathrm{local}\;\mathrm{density}\right)$	$x_{\mathrm{new}}$: newly generated node $x_{\mathrm{near}}$: nearest node $x_{\mathrm{rand}}$: random sample η: adaptive step size *f*: local density function Adapts step size dynamically for dense environments
**Deep Sampling-Based Planner** [81]	$x_{\mathrm{sample}}∼P_{\mathrm{NN}}\left(x|\mathrm{map},\mathrm{goal}\right)$	$N_{NN}$: neural sampling distribution Learns goal-oriented sampling of feasible regions
**Reward-Adaptive Sampling** (extension of [82,83])	$P\left(x_{j}\right)=\frac{w_{j}}{\sum_{i}w_{i}},\;w_{j}=\frac{1}{d_{j}^{\gamma }}\cdot r_{j}$	$P(x_{j})$: adaptive exploration probability $r_{j}$: reward factor Dynamically balances exploration and exploitation
**NAMR-RRT (Neural Adaptive and Multi-Risk RRT)** [80,81,84]	Integrates neural guidance and risk-awareness into RRT*, adapts sampling with dynamic obstacle metrics, and manages uncertainty.	Adapts neural risk-weighted sampling for complex and dynamic regions.

Note: The asterisk “*” in RRT* denotes the optimized variant of the Rapidly-exploring Random Tree algorithm that guarantees asymptotic optimality.

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
