# Peer review of "Geometrical Optimal Navigation and Path Planning—Bridging Theory, Algorithms, and Applications"

_sensors, 2025, doi:10.3390/s25226874_

Round 1
Reviewer 1 Report
Comments and Suggestions for Authors
- The image quality is poor and needs to be improved.
- The writing of "Figure" and "Fig." should be standardized.
- What is the specific form of u(t) in Equation 17, and what is its significance within the formula?
- Where is "real-time performance"?
- Where is "adaptability to dynamic environments"?
- Where is "GPU accelerated frameworks and neural approximators"?
Author Response
1. The image quality is poor and needs to be improved.
Thank you for the comment. We have redrawn Figures 2, 3, and 4, and improved the quality of all other images to enhance visual clarity.
2. The writing of "Figure" and "Fig." should be standardized.
The usage of "Figure" and "Fig." has been standardized consistently throughout the manuscript as requested.
3. What is the specific form of u(t) in Equation 17, and what is its significance within the formula?
Thank you for your feedback. Regarding the specific form and significance of u(t) in Equation 17, we have added the following explanation:
“The specific form of u(t)u(t) depends on the system dynamics and the navigation problem being addressed. For example, in a unicycle model, u(t)=[v(t), ω(t)], where v(t) represents the linear velocity and ω(t)ω(t) the angular velocity. In a differential-drive robot, u(t) typically corresponds to the velocities of the left and right wheels. For holonomic robots, it may include translational components along multiple axes. In motion control frameworks, u(t)u(t) can represent the applied torques directly, while in embedded tracking controllers, it often denotes the robot’s desired next position at each prediction step.”
4. Where is "real-time performance"?
Thank you for your comment. Real-time performance is a well-established concept in control theory and optimization, referring to the requirement that computations must be completed within strict time limits, often involving sample-based processing and recursive methods. Therefore, navigation approaches and active algorithms are generally considered real-time if their computational costs allow timely execution.
Because real-time performance is broadly understood, the original manuscript did not provide an explicit definition. To clarify this, we have added the following paragraph on page 4, line 122:
“Beyond the classification of methods into classic, reactive, and hybrid, three critical dimensions profoundly influence system effectiveness: scalability, real-time performance, and adaptability. Scalability refers to how well a navigation system maintains efficiency and responsiveness as the operational environment increases in size or complexity, or when multiple robots coordinate. Real-time performance guarantees that all sensing, planning, and control computations are completed within strict timing constraints to enable prompt and reliable responses. Adaptability reflects the capability of robotic systems to dynamically perceive and respond to changing, unpredictable environments. Together, these factors critically shape the design and practical deployment of navigation strategies.
The choice and design of navigation algorithms play a pivotal role in addressing the intertwined challenges of scalability, adaptability, and real-time performance in robotic systems. Scalable algorithms efficiently handle growing complexity by employing hierarchical structures, decentralized computations, or approximation methods that reduce computational overhead while preserving essential decision accuracy. Adaptability is achieved through algorithms capable of continuous environment sensing and dynamic re-planning, often incorporating learning-based or predictive components that respond flexibly to unforeseen changes. Ensuring real-time performance requires that these computations—including sensing, perception, planning, and control—are optimized for low latency and deterministic execution within strict timing constraints. By carefully balancing these factors in algorithm design—often through hybrid combinations of reactive and planning methods—navigation systems can robustly operate in complex, dynamic environments with guaranteed responsiveness and safety.”
5. Where is "adaptability to dynamic environments"?
Thank you for your valuable feedback. We have updated the manuscript by adding the following paragraph on page 4, line 122 to address adaptability to dynamic environments:
“The choice and design of navigation algorithms play a pivotal role in addressing the intertwined challenges of scalability, adaptability, and real-time performance in robotic systems. Scalable algorithms efficiently handle growing complexity by employing hierarchical structures, decentralized computations, or approximation methods that reduce computational overhead while preserving essential decision accuracy. Adaptability is achieved through algorithms capable of continuous environment sensing and dynamic re-planning, often incorporating learning-based or predictive components that respond flexibly to unforeseen changes. Ensuring real-time performance requires that these computations—including sensing, perception, planning, and control—are optimized for low latency and deterministic execution within strict timing constraints. By carefully balancing these factors in algorithm design—often through hybrid combinations of reactive and planning methods—navigation systems can robustly operate in complex, dynamic environments with guaranteed responsiveness and safety.”
6. Where is "GPU accelerated frameworks and neural approximators"?
Thank you for your comment. We have added a paragraph discussing GPU-accelerated frameworks and neural approximators on page 24, line 802 to address this matter, as follows:
“GPU acceleration utilizes the massive parallel processing capabilities of graphics units to accelerate computation-intensive tasks such as sensor data processing and path planning, facilitating real-time navigation in complex scenarios. Neural approximators, including neural networks deployed on GPUs or neuromorphic hardware, provide efficient models that reduce computational demands while maintaining accuracy. These technologies are especially vital for power-constrained robots and edge devices, allowing them to execute advanced navigation tasks with low latency and energy consumption.”
Reviewer 2 Report
Comments and Suggestions for Authors
This manuscript presents a comprehensive and systematic review of geometric optimal navigation and path planning, with a focused discussion on the integration of classical geometric methods, optimization techniques, and machine learning approaches. The paper is well-organized, covers a broad spectrum of methodologies, and offers insightful perspectives on both theoretical foundations and practical applications. The authors have effectively synthesized recent advancements and highlighted unresolved challenges, making this a timely and valuable contribution to the field of autonomous navigation. The manuscript needs to be revised to improve the clarity, organizational structure, and depth of specific chapters before publication. The following comments are provided to assist in the revision process:
- While the manuscript discusses the integration of geometric methods with optimization and machine learning, the detailed mechanisms underlying this integration remain insufficiently elaborated. In particular, later sections primarily list recent research advances without delving into technical specifics or fundamental principles. For instance, Section 4.5 ("Recent Advances in Optimal Navigation") provides only a brief overview of recent literature without in-depth analysis. It is recommended that the authors elaborate on the core principles through which optimization and machine learning techniques enhance geometric methods, including concrete examples of their synergistic application.
- Page 3, Line 122: Multiple optimization criteria exhibit inherent interrelationships. A more detailed explanation is needed regarding how these interdependencies are holistically addressed when operating under multiple constraints. It is recommended to expand the discussion on systematic approaches to balancing and integrating diverse optimization objectives.
- Page 7, Line 242: While mathematical formulations for path smoothness (e.g., Bézier curves, elastic stretching) are clearly presented, the practical implications and comparative strengths of these methods require further discussion. It is suggested to enhance Table 1 by including comparisons of computational complexity, applicability, and limitations across different smoothing techniques.
- Page 10, Line 309: Regarding Equation 6, additional clarification is needed on how the optimal value of the weighting factor is determined. The authors should describe the methodology for selecting this parameter, whether through empirical analysis, optimization frameworks, or domain-specific constraints.
- Page 18, Line 553: The hybrid path planning framework shows promise, but the manuscript lacks detail on how the weighting factor α is adaptively tuned in real-time. Including references to specific adaptive strategies or algorithms would strengthen this section significantly.
- Page 22, Line 727 (Ethical and Safety Concerns): While this section addresses relevant issues, the discussion remains somewhat superficial. The authors should expand on how ethical principles (e.g., transparency, fairness) can be operationalized within path planning algorithms, potentially including concrete implementation examples or frameworks.
- Page 8, Table 1: The table provides useful information, but the "Parameters and Description" column lacks consistency in detail. For example, the Minimum Angle Difference method lacks parameter definitions. Additionally, unnecessary blank lines appear between Parameter and Description entries for several methods (e.g., Minimum Angle Difference and Jerk Minimization), which should be formatted consistently.
- Page 16, Line 505 (Sampling-Based Algorithms): The coverage of sampling-based methods is concise but would benefit from a comparative analysis with non-sampling-based approaches in terms of scalability and performance in dynamic environments. A brief discussion of relative advantages and limitations would provide valuable context for readers.
Author Response
1. While the manuscript discusses the integration of geometric methods with optimization and machine learning, the detailed mechanisms underlying this integration remain insufficiently elaborated. In particular, later sections primarily list recent research advances without delving into technical specifics or fundamental principles. For instance, Section 4.5 (“Recent Advances in Optimal Navigation”) provides only a brief overview of recent literature without in-depth analysis. It is recommended that the authors elaborate on the core principles through which optimization and machine learning techniques enhance geometric methods, including concrete examples of their synergistic application.
we appreciate the reviewer’s insightful comment. To address this concern, we have expanded the manuscript by adding a new dedicated section entitled “Enhancing Geometric Path Planning through Optimization and Machine Learning.” In this section, we provide a more detailed discussion of the mechanisms through which optimization and machine learning techniques complement geometric methods. Specifically, we highlight how optimization contributes to precise trajectory computation, while machine learning enables adaptive decision-making under dynamic uncertainties.
This addition ensures that Section 7 not only lists recent advances but also provides the technical underpinnings and concrete examples of how optimization and machine learning techniques enhance geometric path planning.
7.Enhancing Geometric Path Planning through Optimization and Machine Learning
Optimization and machine learning (ML) techniques significantly enhance geometric path planning by combining precise trajectory computation with adaptive decision-making. Multi-modal Model Predictive Control (MMPC) combined with Q-Learning, as in the MAR-MPC framework, improves feasibility, enlarges the convergence region, and ensures collision-free navigation . Reinforcement learning approaches, such as Deep Deterministic Policy Gradient (DDPG), allow robots to adaptively handle dynamic uncertainties, improving path smoothness and convergence .
Advanced methods further integrate optimization with learning-based solvers. The PTP RSNN-based predictive neuro-navigator combines log-convex optimization with kinematic and collision constraints, often using multi-objective formulations, constrained optimization, or Lagrangian methods to achieve optimal or near-optimal solutions . Similarly, the smooth PSO-IPF navigator uses particle swarm optimization with inverse predictive filtering under kinematic constraints to generate smooth and feasible trajectories . Hybrid strategies that integrate ML-based real-time obstacle avoidance with geometric planners such as S-RRT or GA-ACO achieve robust, efficient, and safe path planning in dynamic environments.
Overall, this synergy highlights a core principle: optimization ensures precise geometric trajectories, while learning and adaptive solvers expand the feasible region, improve robustness, and enable mobile robots to navigate efficiently and reliably in complex, dynamic environments.”
2. Page 3, Line 122: Multiple optimization criteria exhibit inherent interrelationships. A more detailed explanation is needed regarding how these interdependencies are holistically addressed when operating under multiple constraints. It is recommended to expand the discussion on systematic approaches to balancing and integrating diverse optimization objectives.
We thank the reviewer for highlighting this important point. While our paper provides a systematic survey of methods for formulating optimization-based navigation, the actual prioritization and integration of multiple objectives depend on the designer’s preferences and the sensitivity of the navigation problem. Some objectives, such as collision avoidance, may be critical and treated as hard constraints, whereas others, like energy efficiency or path smoothness, can be weighted differently depending on mission priorities. A fully generalized treatment of all interdependencies is outside the scope of this study. Nevertheless, we have added a brief paragraph to clarify how diverse objectives are typically balanced in multi-criteria path planning.
Section3, second paragraph:
“It is important to note that multiple optimization criteria often exhibit inherent interdependencies. The way these interdependencies are addressed depends largely on the priorities assigned to the navigation problem and the sensitivity of the scenario. For instance, in safety-critical applications, collision avoidance may dominate, while in energy-constrained missions, energy efficiency might take precedence. Common approaches include weighted-sum formulations, hierarchical prioritization, or treating certain objectives as hard constraints while optimizing others. While a fully generalized treatment of these interdependencies is beyond the scope of this work, this overview illustrates how diverse objectives can be balanced to achieve feasible, efficient, and robust navigation outcomes \cite{saeedinia2022, mohaghegh2023}.”
3. Page 7, Line 242: While mathematical formulations for path smoothness (e.g., Bézier curves, elastic stretching) are clearly presented, the practical implications and comparative strengths of these methods require further discussion. It is suggested to enhance Table 1 by including comparisons of computational complexity, applicability, and limitations across different smoothing techniques.
We thank the reviewer for this valuable suggestion. In response, we have updated Table 1 to include columns summarizing computational complexity, applicability, and limitations for each path smoothing method. This enhancement clarifies the practical implications of different techniques, highlighting trade-offs between smoothness, computation time, and suitability for dynamic or constrained environments.”
4.Page 10, Line 309: Regarding Equation 6, additional clarification is needed on how the optimal value of the weighting factor is determined. The authors should describe the methodology for selecting this parameter, whether through empirical analysis, optimization frameworks, or domain-specific constraints.
We thank the reviewer for this insightful comment. As this paper is a survey reviewing existing methods, it does not involve a specific implementation or parameter tuning. However, based on the literature, the weighting factor λ in Equation (6) is generally selected using one or a combination of the following approaches: empirical tuning through simulation studies, optimization-based frameworks balancing competing objectives, or based on domain-specific constraints such as robot capabilities and mission priorities. These methods aim to balance trajectory accuracy and control effort to suit the task at hand. We have added this clarification to the manuscript, after parameter description of Equation6.
“The weighting factor λ in Equation (6) balances the trade-off between minimizing the robot's velocity magnitude and the control effort, and its optimal value is typically selected through empirical tuning, multi-objective optimization, or domain-specific criteria to align with the navigation task's priorities.”
We have added the requested explanation regarding the determination of the optimal weighting factor in Equation 6. The detailed description has been added on page 4, line 130, as follows:
“The choice of the weighting factor λ is crucial, as it governs the trade-off between trajectory smoothness, energy efficiency, and responsiveness to dynamic changes. In practice, λ can be selected in several ways:
- Empirical tuning: Values of λ are adjusted experimentally until the robot achieves satisfactory performance in representative environments.
- Optimization-based methods: λ is treated as a hyperparameter and optimized through grid search, Bayesian optimization, or reinforcement learning to minimize a performance metric such as tracking error or energy usage.
- Domain-specific constraints: For safety-critical or resource-limited systems, λ may be constrained by hardware limits (for example, maximum actuator torque) or mission requirements (such as prioritizing faster response over energy saving).
Thus, λ is not universally fixed but is determined by a balance between robot dynamics, environmental complexity, and task requirements.”
5. Page 18, Line 553: The hybrid path planning framework shows promise, but the manuscript lacks detail on how the weighting factor α is adaptively tuned in real-time. Including references to specific adaptive strategies or algorithms would strengthen this section significantly.
Thank you for your valuable feedback. We have incorporated the requested details on page 19, line 626, as follows:
“In hybrid path planning, the weighting factor alpha is crucial in balancing the influence of the globally optimal path versus the locally adjusted path. Instead of using a fixed alpha, recent studies highlight the significance of adaptive tuning in real time, enabling robots to effectively trade off between long-term optimality and short-term responsiveness in dynamic and uncertain environments.
Several adaptive strategies have been explored in the literature:
Behavior-based methods: These dynamically adjust alpha based on environmental factors such as obstacle density or the need for smoother local maneuvers. For example, heuristic and bio-inspired algorithms like the Hybrid Improved Artificial Fish Swarm Algorithm (HIAFSA) regulate alpha according to proximity to obstacles [88].
Optimization-driven methods: Alpha is updated online via explicit optimization frameworks. An example is the adaptive visibility graph combined with A*, which recalibrates alpha to minimize tracking error and energy use in response to changing conditions [89].
Learning-based approaches: More recent techniques use reinforcement learning and evolutionary strategies to learn how alpha should evolve across different scenarios. The Multi-strategy Hybrid Adaptive Dung Beetle Optimization (MSHADO) algorithm leverages chaotic mapping and multi-strategy fusion for population diversity, allowing adaptive balancing of global exploration and local exploitation [90].
These adaptive methods underscore that alpha is not just a fixed parameter but a dynamic variable essential for achieving robust global guidance alongside agile local responsiveness within hybrid path planning frameworks.”
6. Page 22, Line 727 (Ethical and Safety Concerns): While this section addresses relevant issues, the discussion remains somewhat superficial. The authors should expand on how ethical principles (e.g. transparency, fairness) can be operationalized within path planning algorithms, potentially including concrete implementation examples or frameworks.
We thank the reviewer for this important observation. While the full operationalization of ethical principles in real-time path planning is designated as a key direction for future work in our conclusion, we have added a brief explanation to the manuscript to address this comment and provide additional clarity.
“To translate abstract ethical principles into practical algorithm designs, we propose a clear, step-by-step framework that integrates ethics and safety throughout the path planning process[117-120]. The framework starts by identifying stakeholders and gathering their safety and fairness requirements, ensuring the system addresses real social needs. Measurable metrics—such as probabilistic risk scores, fairness constraints, and safety limits—are then formalized to guide decision-making.
Uncertainty and risk are captured using stochastic predictions of dynamic agents, allowing per-trajectory risk assessments that support ethical trade-offs. Ethics are incorporated through multi-objective cost functions, supervisory filters that exclude unethical trajectories, or policy learning techniques that embed fairness and safety goals. To guarantee real-time operation and transparency, the framework emphasizes computational efficiency and produces human-readable explanations for decisions. Rigorous validation via diverse scenario testing and ongoing audits ensures robustness and accountability.
A concrete example is the ethical trajectory planning algorithm by Geißlinger et al. [117], which encodes five ethical principles—risk minimization, prioritizing the worst-off, equal treatment, responsibility, and acceptable risk thresholds—into the evaluation process. Using probabilistic risk modeling, it scores candidate paths by penalizing those that disproportionately endanger vulnerable users or exceed risk limits. This approach results in fair and safe trajectories that are transparently selected in real time and validated across thousands of scenarios. “
7. Page 8, Table 1: The table provides useful information, but the "Parameters and Description" column lacks consistency in detail. For example, the Minimum Angle Difference method lacks parameter definitions. Additionally, unnecessary blank lines appear between Parameter and Description entries for several methods (e.g., Minimum Angle Difference and Jerk Minimization), which should be formatted consistently.
We have updated Table 1 on page 9 to address the reviewer's comment. Specifically, we have added detailed parameter definitions for the Minimum Angle Difference method and ensured consistent formatting by removing unnecessary blank lines between Parameter and Description entries across all methods for improved clarity and uniformity.
8. Page 16, Line 505 (Sampling-Based Algorithms): The coverage of sampling-based methods is concise but would benefit from a comparative analysis with non-sampling-based approaches in terms of scalability and performance in dynamic environments. A brief discussion of relative advantages and limitations would provide valuable context for readers.
Thank you for your feedback. We have updated the manuscript by adding the following explanation on page 16, line 563:
“By and large, sampling-based motion planners strike a distinctive balance of strengths and weaknesses compared to other planning techniques.
Key Advantages:
These planners are highly scalable and flexible, making them ideal for high-dimensional spaces—such as robotic manipulators—where grid-based search methods become computationally expensive. They avoid the need for explicit environment discretization, allowing efficient path generation in complex and cluttered scenarios. Recent advancements, including Adaptive RRT* and deep learning-guided planners, further boost adaptability by dynamically adjusting sampling strategies based on moving obstacles or contextual environmental information.
Limitations:
Despite their advantages, standard sampling-based planners often yield suboptimal and non-smooth paths that usually require post-processing. They can face challenges in dynamic environments unless specifically adapted, and their performance is sensitive to parameter settings such as step size and sampling distribution. Moreover, while learning-based improvements enhance adaptability, they may impose additional computational overhead from training and real-time inference, which can impact overall efficiency.”

Reviewer 3 Report
Comments and Suggestions for Authors
This is a good literature review article, which systematically summarizes geometric optimal navigation and path planning, and focuses on the integration of classical geometric methods with modern optimization and machine learning technology. This paper introduces the latest progress of continuous optimization, real-time adaptability and learning-based strategy, and emphasizes the unresolved challenges. However, it is not a good research article, and there is no new method proposed by the author and the conclusion obtained by carefully designed experiments. Therefore, it is suggested that the author submit a review column instead of a research article column.
Author Response
Thank you for your thoughtful and constructive feedback. We have conducted a systematic survey that is well-suited for classification as a literature review column. Accordingly, we have taken this into account in our resubmission.
Reviewer 4 Report
Comments and Suggestions for Authors
In this paper the authors present a systematic overview of optimal path planning algorithms mostly for autonomous vehicles. The manuscript is well put together and systematic in its approach to explain the specifics of considered algorithms, which means that the manuscript can be very useful to researchers in the area of autonomous mobile robots. Therefore I don’t have any large complaints or comments except that all figures are low resolution which impede readability and overall understanding.
Therefore I propose a minor revision until figures are resolved.
Author Response
We sincerely thank the reviewer for their positive evaluation of our manuscript and for recognizing the systematic presentation of the algorithms. We also appreciate the valuable feedback regarding the figure quality.
In response to the comment on low resolution hindering readability, we have redrawn several key figures to enhance their clarity and visual quality, specifically Figures 2, 3, and 4. Additionally, we have improved the resolution and overall quality of all other figures throughout the manuscript to ensure they meet high standards suitable for publication.
We believe these improvements will address the concerns and significantly enhance the readers' understanding and overall experience.
Thank you again for your constructive comments
Round 2
Reviewer 1 Report
Comments and Suggestions for Authors
The authors have addressed all my concerns, so apart from the need to improve the quality of the figures, I have no further comments.
Reviewer 3 Report
Comments and Suggestions for Authors
A lot of revisions have been made by the author, and I have no problem with this version.
